# MEMFORMER: THE MEMORY-AUGMENTED TRANSFORMER

## ABSTRACT

Transformer models have obtained remarkable accomplishments in various NLP tasks. However, these models have efficiency issues on long sequences, as the complexity of their self-attention module scales quadratically with the sequence length. To remedy the limitation, we present Memformer, a novel language model that utilizes a single unified memory to encode and retrieve past information. It includes a new optimization scheme, Memory Replay Back-Propagation, which promotes long-range back-propagation through time with a significantly reduced memory requirement. Memformer achieves $\mathcal{O}(n)$ time complexity and $\mathcal{O}(1)$ space complexity in processing long sequences, meaning that the model can handle an infinite length sequence during inference. Our model is also compatible with other self-supervised tasks to further improve the performance on language modeling. Experimental results show that Memformer outperforms the previous long-range sequence models on WikiText-103, including Transformer-XL and Compressive Transformer.

## 1 INTRODUCTION

Memory has a fundamental role in human cognition. Humans perceive and encode sensory information into a compressed representation in neurons, and later our brains can effectively retrieve past information to accomplish various tasks. The formation of memories involves complex cognitive processes. Modeling and studying the behavior of human memory is still a challenging research problem in many academic areas.

Many researchers have attempted to incorporate memory systems in artificial neural networks. Early works like recurrent neural networks (RNN) (Rumelhart et al., 1988), including LSTM (Hochreiter & Schmidhuber, 1997) model temporal sequences with their internal compressed state vector as memory. Although RNNs are theoretically Turing-complete, they are limited in preserving the long-term information due to the memory bottleneck. To alleviate the limitation, more powerful memory network architectures such as Neural Turing Machine (NTM) (Graves et al., 2014), Differential Neural Computer (DNC) (Graves et al., 2016) have been proposed by leveraging a large external memory. However, due to their complex memory addressing mechanism, they are not widely used in NLP.

More recently, Vaswani et al. (2017) proposes Transformer by ditching the use of memory and recurrence. Instead, it maintains all $\mathcal{O}(N^2)$ dependencies in the sequence with self-attention (Bahdanau et al., 2015). Transformer and its followers have achieved great success in various NLP tasks. Nevertheless, the quadratic complexity can be extremely costly when the input sequence is long. Some works address the limitations of self-attention, including Reformer, Sparse Transformer, Longformer, Linformer, etc (Child et al., 2019; Kitaev et al., 2020; Wang et al., 2020). They successfully reduce the complexity of self-attention and can process longer sequences. However, the space cost still scales with sequence length, which cannot be fully eliminated without memory and recurrence.

Transformer-XL (Dai et al., 2019) re-introduces the concept of memory and recurrence. It caches each layer's hidden states of self-attention into a fixed size queue and re-uses them in the later attention computation. However, the memory as raw hidden states cannot effectively compress high-level information. Transformer-XL in practice needs a huge memory size to perform well. Compressive

Transformer (Rae et al., 2020) improves upon Transformer-XL by further compressing its memories into fewer vectors via a compression network. However, as mentioned in the papers, both Transformer-XL and Compressive Transformer still have a theoretical maximum temporal range due to the uni-directional self-attention constraint.

In this work, we propose Memformer, which includes a more efficient memory system with a Transformer encoder-decoder architecture. The resulting model has a theoretically unlimited temporal range of memorization. We also improve the relative positional encoding in Transformer-XL with a simplified version. As the traditional back-propagation through time (BPTT) has an unaffordable memory cost for our model, we introduce a new optimization scheme, memory replay back-propagation (MRBP), to significantly reduce the memory cost of training recurrent neural networks with large memory. We show that Memformer is compatible with different self-supervised tasks and can further improve its performance on language modeling.

Our main contributions can be summarized as follows: (1) We introduce a new optimization scheme for training recurrent neural networks with large memory and long temporal range. (2) We propose Memformer, a Transformer-based model, which outperforms the previous Transformer-XL and Compressive Transformer on WikiText-103 language modeling. (3) We show that Memformer is compatible with a wide range of self-supervised tasks other than autoregressive language modeling.

## 2 METHODS

### 2.1 SIMPLIFIED RELATIVE POSITIONAL ENCODING

The standard attention mechanism involves the dot product between the query vector $q_i$ and the key vector $k_j$, where $\mathbf{W}_q, \mathbf{W}_k, \mathbf{W}_v$ are the projection matrices to produce the query, key, and value. TransformerXL proposes a new type of relative positional encoding method. The attention computation is decomposed into four parts: $(a)$ content-based addressing, $(b)$ content dependent positional bias, $(c)$ global content bias, and $(d)$ global positional bias. The relative positional embedding $R_{i-j}$ provides the positional information between every pair of $x_i$ and $x_j$. The equation is defined below. $u$ and $v$ are trainable parameters.

$$\mathbf{A}_{i,j} = \underbrace{\mathbf{E}_{x_i}^\top \mathbf{W}_q^\top \mathbf{W}_r \mathbf{E}_{x_j}}_{(a)} + \underbrace{\mathbf{E}_{x_i}^\top \mathbf{W}_q^\top \mathbf{W}_r R_{i-j}}_{(b)} + \underbrace{u^\top \mathbf{W}_k \mathbf{E}_{x_j}}_{(c)} + \underbrace{v^\top \mathbf{W}_r R_{i-j}}_{(d)}. \tag{1}$$

However, we observe that $(c)$ and $(d)$ can be simplified by introducing a bias term to the original query and key projection. Thus, we re-formalize the self-attention, as shown in Eq. 3. The product of $b_q$ and $\mathbf{K}_x$ is equivalent to the term $(c)$ global content bias. For the term $(d)$, since $v$, $\mathbf{W}_r$, and $R_{i-j}$ are all trainable parameters, it can be simplified into the product between $b_q$ and $b_k$, which has a similar effect to the global attention bias. Different from Transformer-XL that only injects positional information in the attention computation, our attention mechanism shown in Eq. 4 attends over the positional information and accumulate the results to have more robust output representations.

$$\mathbf{Q}_x = \mathbf{W}_q \mathbf{E}_x + b_q; \quad \mathbf{K}_x = \mathbf{W}_k \mathbf{E}_x + b_k; \quad \mathbf{V}_x = \mathbf{W}_v \mathbf{E}_x + b_v \tag{2}$$

$$\mathbf{A}_{i,j} = \mathbf{Q}_{x_i}^\top \mathbf{K}_{x_j} + \mathbf{Q}_{x_i}^\top \mathbf{R}_{i-j} \tag{3}$$

$$\mathbf{H}_x = \sum_j \mathbf{A}_{i,j} \left( \mathbf{V}_{x_j} + R_{i-j} \right) \tag{4}$$

### 2.2 MEMFORMER

This section explains the details of Memformer. We first talk about the language model background and a new way of formulating language generation with text continuation. Then we describe an instance of such formulation, which is our proposed Memformer model. After that, we introduce the multi-task training setting. Finally, we describe the newly proposed optimization scheme, memory reply back-propagation to tackle the memory cost problem.

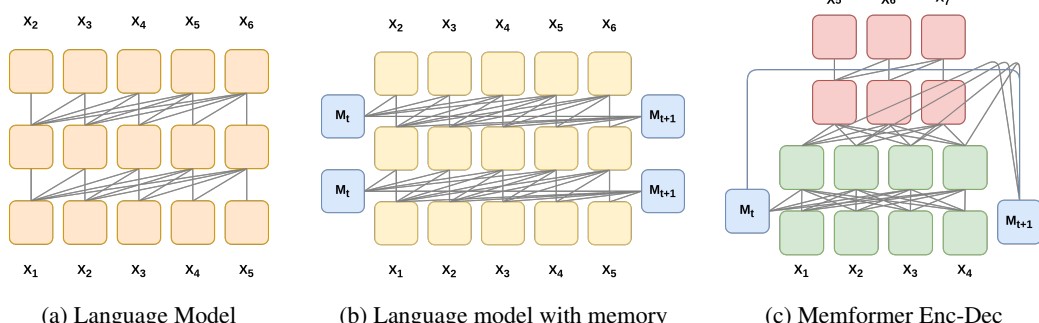

(a) Language Model    (b) Language model with memory    (c) Memformer Enc-Dec

Figure 1: Illustrations of different language model architectures. (a) is the standard Transformer decoder. (b) is a language model with memory. The decoder takes $M_t$ and outputs the next timestep's memory $M_{t+1}$ at the end of segment. (c) is a Transformer encoder-decoder model. The encoder is modified to read and write memory.

### 2.2.1 BACKGROUND: STANDARD LANGUAGE MODEL

To understand Memformer better, we first study the standard language model. Given a document of $N$ tokens $x = (x_1, x_2, \ldots, x_N)$, an standard language model learns the joint probability of the document by taking the product of each token's probability conditioned to the previous tokens, which is defined as $P(x) = \prod_t P(x_t|x_{1:t})$.

Figure 1a and 1b are the standard language models. They autoregressively predict the next token by feeding the previous generated tokens into the model. An extension of Figure 1a is to incorporate relative positional encoding and cache the past hidden states. Then this model would be equivalent to Transformer-XL.

Figure 1b is an assumed language model with memory. Self-attention module now attends not only to its token inputs but also to the memory $M_t$ at time $t$. After all the tokens in the segment are processed, the model summarizes the computed hidden states in the segment and produce the next timestep's memory $M_{t+1}$. Each layer has its own individual memory representation. One limitation for this model is that the read and write operations on memory may not have enough capacity to retain important information due to the uni-directional attention.

### 2.2.2 ENCODER-DECODER LANGUAGE MODEL

To address this capacity issue of uni-directional attention, we introduce a more powerful architecture shown in Figure 1c, where we have an encoder-decoder and a memory system. If a document is split into $T$ segments of length $L$, for each segment $s_t$, we define $s_t = [x_{t,1}, x_{t,2}, \ldots x_{t,L}]$. The encoder's role is to encode the segment $s_t$ and inject the information into the memory $M_t$, while it also retrieves past information from the previous timestep's memory $M_{t-1}$. The final output of the encoder will be fed into the decoder's cross attention layers to predict the token probabilities of the next timestep's segment $s_{t+1}$ as standard language modeling. The definition is as below:

$$M_t = \text{Encoder}(s_t, M_{t-1}) \tag{5}$$

$$P(s_t) = \prod_{n=1:L} P_{\text{Decoder}}(x_{t,n} \mid x_{t,<n}, M_{t-1}) \tag{6}$$

$$P(x) = \prod_{t=1:T} P_{\text{Model}}(s_t|s_{<t}) \tag{7}$$

At each timestep, the process can be deemed as a text continuation task. Given a text segment as the input, the model needs to continue that segment by generating the next text segment. Since the memory stores all the past information, we can autoregressively generate all the text segments in a document. In this fashion, the model can behave as a language model.

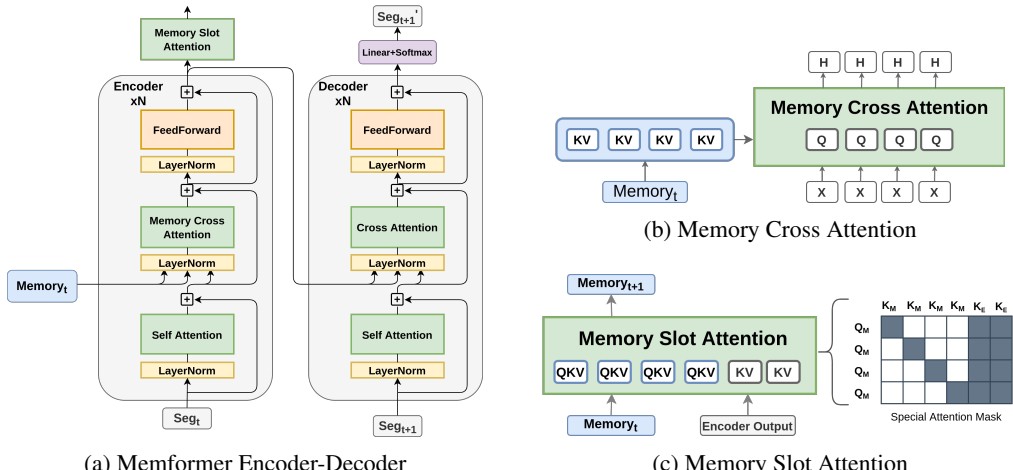

Figure 2: Illustration of Memformer Encoder-Decoder and its sub-modules. (a) shows the overall architecture of Memformer Encoder-Decoder. (b) demonstrates the memory cross attention for reading from memory. (c) is the memory slot attention module to write the next timestep's memory.

### 2.2.3 MEMFORMER ENCODER-DECODER

To implement the encoder-decoder language model, we propose Memformer Encoder-Decoder. The model incorporates a Transformer encoder-decoder and a memory system. The encoder is equipped with two new modules: Memory Cross Attention (Figure 2b) and Memory Slot Attention (Figure 2c) to read from or write to the memory respectively. The encoder is fully responsible for encoding and retrieving past information via memory. The decoder then takes the last layer's outputs from the encoder and feeds them into the cross attention module similar to the standard Transformer. For the text continuation task, we let the encoder take the input of the current timestep's text segment, and let the decoder generate the next timestep's segment tokens. Figure 2a shows the detailed structure.

Figure 2b demonstrates how Memory Cross Attention module extracts information from the memory $M_t$ with the current segment's tokens $X$. Each input token's hidden state is projected into queries, while the memory hidden states are projected into key-value pairs. Then the input hidden states will attend over the projected memory key-value pairs to produce the final outputs. This module can effectively retrieve past information from memory $M_t$ given the current text segment.

Memory Slot Attention in Figure 2c produces the next timestep's memory $M_{t+1}$. This module takes the inputs of the previous timestep's memory $M_t$ and the encoder's final hidden states. It then projects the memory into queries, keys, and values, while the encoder outputs are into keys and values. Since each memory slot should not be interfering with other memory slots, we design a special type of sparse attention pattern (details shown in Figure 2c). Thus, each slot in the memory can only attend over itself and the encoder outputs. This is to preserve the information in each slot longer over the time horizon. For example, if one slot only attends itself, then the information in that slot will not change in the next timestep.

### 2.3 MULTI-TASK SELF-SUPERVISED LEARNING

Unlike existing models built either for denoising objectives or language modeling, Memformer can accomplish both types of tasks. This flexibility helps the model learn better representations of the document and strengthen the memory of past information. To avoid conflicts of different tasks, we use separate special tokens for each task. In this work, we only experiment with three self-supervised tasks. We believe that our model is flexible with many other self-supervised tasks to further improve performance. We randomly sample the following three tasks with a probability $[0.6, 0.3, 0.1]$ during training.

**Text Continuation** This is the primary task, as our goal is for language modeling. Given the current timestep's text segment, the model needs to generate the tokens in the next timestep's segment.

**Text Infilling**    This task is inspired by BART (Lewis et al., 2020). We mask some text spans in a document. The span length is drawn from a Poisson distribution ($\lambda = 3.5$). The span is replaced with a "[mask]" token. The model needs to predict these masked tokens.

**Text Recall**    Reverse of the text continuation task, Text Recall needs to predict the previous text segment given the current timestep's segment. This task aims to directly help the model to better preserve the past information.

## 2.4 MEMORY REPLAY BACK-PROPAGATION

Memformer relies on the explicit memory to encode long-range document. At inference time, there is no additional memory cost because of the single unified memory design. Nevertheless, during training, such design would require back-propagation through time (BPTT) over a long range of timesteps so that the memory writer network can potentially learn to retain long-term information. The problem with BPTT is that it unrolls the entire computational graph during the forward pass and stores all the intermediate activations. This process would lead to impractically huge memory consumption for Memformer, which causes training almost impossible.

A favorable existing approach to eliminate this problem is gradient checkpointing (Chen et al., 2016). The algorithm can significantly reduce the memory cost of a large computational graph. However, the standard gradient checkpointing still needs to compute all the nodes in the computational graph and store unnecessary hidden states during the forward pass. We propose Memory Replay Back-Propagation (MRBP), a more efficient variant of gradient checkpointing, by replaying the memory at each timestep to accomplish gradient back-propagation over long unrolls.

MRBP is designed specifically for recurrent neural networks. The algorithm takes an input with a rollout $[x_0, x_1, \ldots, x_T]$ with length $T$ and the previous memory $M_0$. MRBP only traverses the critical path in the computational graph during the forward pass. It then obtains each timestep's memory and stores those memories in the replay buffer. During the backward pass, MRBP backtracks the memories in the replay buffer from time $T$ to $0$ and recompute the partial computational graph for the local timestep. It continues the computation of the remaining graph with the output $O_t$ to get the loss for back-propagation. There are two directions of gradients for the model. One direction of gradients comes from the local back-propagation of loss, while the other part comes from the back-propagation of the next memory's Jacobin $\nabla M_{t+1}$. The full algorithm is described in Algorithm 14

---

**Algorithm 1:** Memory Replay Back-Propagation

**Input:** rollout=$[x_0, x_1, \ldots, x_T]$: a list containing each timestep $t$'s input $x_t$
        prevMemory: memory from the previous rollout

  ▷ initialize a list to store all the memories computed
1 replayBuffer = []
2 replayBuffer.append($M_0$) ;                                                  ▷ previous memory
  ▷ forward pass
3 **for** $t = 0, 1, 2, \ldots, T-1$ **do**
4    |  $M_{t+1},\_ = \text{Model}(x_t, M_t)$ ;                                  ▷ No gradient
5    |  replayBuffer.append($M_{t+1}$)
6 **end**
  ▷ backward pass
7 $\nabla M_{t+1} = 0$
8 **for** $t = T, T-1, \ldots, 1, 0$ **do**
9    |  $M_{t+1}, O_t = \text{Model}(x_t, M_t)$ ;                                ▷ Recompute
10   |  $loss = f_{loss}(O_t)$
11   |  $loss$.backward()
12   |  $M_{t+1}$.backward($\nabla M_{t+1}$) ;                                  ▷ Computes $\nabla M_t$
13 **end**
14 save $M_T$ for next rollout's update

---

## 2.5 TEMPORAL RANGE ANALYSIS

We analyze the theoretical maximum temporal range here. Transformer-XL and Compressive Transformer store the past hidden states in a FIFO queue as their memories. However, they have a theoretical limitation for the maximum temporal range when modeling a sequence. Transformer-XL has a maximum temporal range of $N_m \times L$, where $N_m$ is the memory size, and $L$ is the number of layers. Compressive Transformer extends the temporal range to $L \times (N_m + c \times N_{cm}$, by compressing the memories in Transformer-XL into the new compressed memories with a size of $N_{cm}$ and a compression ratio $c$. If a sequence is longer than the maximum temporal range, the model will lose information when the stored memories are discarded. In contrast, Memformer has a single unified memory system, which theoretically has a maximum temporal range of infinity.

## 3 EXPERIMENTS

### 3.1 SETTINGS

We conduct all experiments on WikiText-103 (Merity et al., 2017), which is a popular long-range language modeling benchmark. It contains 28K articles with an average length of 3.6K tokens per article. We adopt byte pair encoding (BPE) (Sennrich et al., 2016) to avoid unknown tokens. The original Transformer-XL and Compressive Transformer set the attention length to 1,600, close to the average length per article. We argue that such large attention length is against the purpose of using memory to compress information. Therefore, to better demonstrate memory efficiency, we set the input size to 64, and restrict the memory size under 128.

Besides, we make the following changes to the baselines. We have two Transformer-XL model: base and large. Transformer-XL base has 12 layers. Transformer-XL large has 16 layers. Since Compressive Transformer does not have code released, we re-implement the model following the paper. Our Compressive Transformer has 12 layers. The compression ratio is set to 4. For a fair comparison, Memformer Encoder-Decoder has a 4-layer encoder and an 8-layer decoder. For all baselines and our models, the hidden size is set to 512, the feed-forward hidden size to 2048, the number of heads to 8, and the head size to 64. We disable dropout as it causes high variance in the final score due to randomness, making fair comparisons impossible. We use the simplified relative positional encoding for all models as it generally performs better under our setting.

### 3.2 MAIN RESULTS

| Model | Memory Size | #Params | Speed (it/s) | Perplexity ↓ |
|---|---|---|---|---|
| Transformer-XL base | 32 | 66.69M | 1,136 | 23.52 |
| Transformer-XL base | 64 | 66.69M | 928 | 22.44 |
| Transformer-XL base | 128 | 66.69M | 805 | 21.49 |
| Transformer-XL large | 64 | 80.35M | 745 | 21.95 |
| Transformer-XL large | 128 | 80.35M | 640 | 21.25 |
| Compressive Transformer | 16(16) | 79.29M | 746 | 22.79 |
| Compressive Transformer | 32(32) | 79.29M | 702 | 21.98 |
| Memformer Encoder-Decoder | 32 | 76.21M | 994 | 21.26 |
| + Text Infilling | 32 | 76.21M | 994 | 21.19 |
| + Text Recall | 32 | 76.21M | 994 | 21.53 |
| + All Multi-task | 32 | 76.21M | 994 | **21.11** |

Table 1: Results on WikiText-103. "16(16)" means that the XL memory size is 16 and the compressed memory size is 16.

Table 1 shows the results on WikiText-103. We report the number of parameters, the number of items per second as the training speed, and perplexity for a comprehensive comparison. Memformer Encoder-Decoder achieves the best perplexity score with a efficient computation and memory trade-off.

When increasing Transformer-XL's memory size, we observe that the perplexity drops as expected, because the attention length is also increased. Note that the speed decreases with a larger memory size. After we have enlarged the memory size of Transformer-XL to 128, the perplexity is still worse than Memformer Encoder-Decoder, and the speed is much slower. Since Memformer Encoder-Decoder has slightly more parameters, we compare our model with Transformer-XL large, which has 16 layers. In Transformer-XL, the number of layers is important for performance, as the maximum temporal range scales with the number of layers. Transformer-XL large indeed obtains better perplexity scores than Transformer XL-base models. However, our model still achieves better perplexity. Not to mention that Memformer Encoder-Decoder is 55% faster than Transformer-XL large. This suggests that Memformer Encoder-Decoder is more efficient in modeling the document than Transformer-XL.

Compressive Transformer is another baseline we report in the table. It introduces an extra compression network to compress the memory hidden states in Transformer-XL. For a fair comparison, Compressive Transformer has half of the memory size for the compressed memory. With the same memory budget, Compressive Transformer performs better than Transformer-XL. However, the extra compression network requires more number of parameters and computation. We actually find that Transformer-XL is more efficient in terms of the number of parameters and speed under our setting.

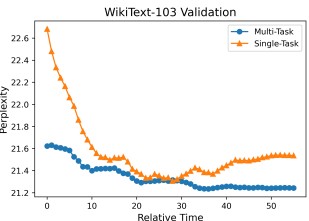 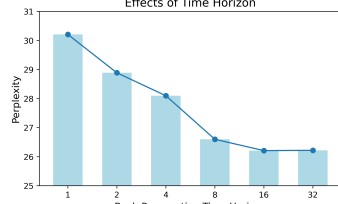 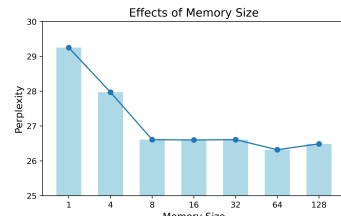

(a) Multi-task learning.      (b) Effects of different time horizons (c) Effects of different memory sizes

Figure 3: Effects of different configurations. (a) shows how multi-task learning helps. (b) shows the effects of changing time horizon. (c) shows the effects of changing memory size.

### 3.3 ABLATION STUDY

We conduct ablation studies to explore how each component contributes to Memformer's good performance, by analyzing the performance improvement of the simplified relative positional encoding and memory replay back-propagation

### 3.4 MODEL HYPER-PARAMETERS

**Effects of Multi-Task Training**     When we combine the three tasks text continuation, text infilling and text recall, the model yields the best performance. We find that when applying only text continuation and text recall, the performance drops. This drop might be because the model over-fits on the text recall task, which hurts the performance of text continuation task. Overall, the performance improvement of multi-task learning is marginal. However, in Figure 3a, we observe that models trained with multi-task learning have a smoother validation curve and is less prone to over-fitting. This indicates that with multi-task learning, the model is more robust and potentially learns better feature representations.

**Effects of Time Horizon**     We test how the time horizon for back-propagation affects the performance. The results are shown in Figure 3b. We vary the back-propagation time horizon from 1 to 32. When the time horizon is set to 1, back-propagation cannot pass gradients through memory to the previous timestep. Thus, we observe the performance is the worst when the time horizon is 1. As we increase the time horizon, the model achieves better perplexity scores. When the time horizon is increased to 32, we observe the marginal improvement on perplexity is almost gone.

**Effects of Memory Size**     A large memory size ideally helps to store more information. From Table 3c, we can see a huge improvement when increasing the memory size from 1 to 8. However, when we further increase the memory size, the perplexity stops decreasing. In future work, we will study how to gain more improvement with larger memory sizes.

### 3.4.1 SIMPLIFIED RELATIVE POSITIONAL ENCODING

To test the performance of our simplified relative positional encoding (RPE), we only replace the self-attention layers in the original Transformer with the new module without changing other parts. The results in Table 2 show that our proposed simplified relative positional encoding has much better performance than the original Transformer-XL's RPE in all metrics.

| Method | Time(ms) | GFLOPS | Parameters | Perplexity |
|--------|----------|--------|------------|------------|
| XL RPE | 1.27 | 68.99 | 1.32M | 23.32 |
| Ours | 0.88 | 68.72 | 1.05M | 21.74 |

Table 2: Comparison for relative positional encoding. The experiments are conducted on Transformer-XL.

| Method | GPU Memory (MB) | Speed (relative) |
|--------|-----------------|------------------|
| BPTT | 16,177 | x1.00 |
| GC | 9,885 | x0.48 |
| MRBP | 7,229 | x0.90 |

Table 3: Memory Replay Back-Propagation performance comparison.

### 3.4.2 MEMORY REPLAY BACK-PROPAGATION

To test MRBP's effectiveness, we compare against the standard back-propagation through time (BPTT) and the standard gradient checkpointing (GC) algorithm. We use Memformer Decoder with 12 layers, 8 heads, 512 hidden size, and 32 memory size for all the experiments here. The time horizon for each truncated back-propagation update is set to 4.

The back-propagation through time (BPTT) approach is the fastest because it does not need re-computation However, it costs the most amount of memory due to unrolling the entire computational graph. While gradient checkpointing can save huge amount of memory, it is much slower than the other two methods (x0.48). In contrast, our MRBP saves more GPU memory with only slight speed degeneration (x0.90). When further increasing the time horizon to 16, we see that the GPU memory only increases 62 MB, suggesting the sub-linear memory growth with the time horizon.

## 4 RELATED WORK

Optimizing the attention pattern of Transformer is one direction to process long sequences. Child et al. (2019) first proposes Sparse Transformer to reduce the computation complexity $\mathcal{O}(N)$ to with a sparse attention pattern for sequence modeling. Longformer (Beltagy et al., 2020) and Big Bird (Zaheer et al., 2020) follow Sparse Transformer and explore the effectiveness of different sparsity patterns. Reformer (Kitaev et al., 2020) applies a multi-round locality-sensitive hashing (LSH) to reduce the computation complexity to $\mathcal{O}(N \log N)$. Linformer (Wang et al., 2020) further reduces the complexity to $\mathcal{O}(N)$ by observing that self-attention is low-rank and can be approximated with linear attention. However, the memory cost of these approaches still scales with the sequence length.

Meanwhile, applying recurrence to Transformers is an orthogonal direction comparing to the efficient attention approaches. Recurrence enables the model to have constant memory complexity $\mathcal{O}(1)$ during inference. There are mainly two works exploring this direction. TransformerXL (Dai et al., 2019) uses relative positional encoding and consists of a segment-level recurrence mechanism to encode beyond a fixed-length context. Compressive Transformer (Rae et al., 2020) extends from Transformer XL by further compressing the previous segment information to achieve longer context. However, they have a theoretical maximum temporal range of context related to the memory size and the number of layers. In practice, TransformerXL and Compressive Transformer needs huge memory size to achieve good performance and are inefficient in their memory representations.

## 5 CONCLUSION

In this work, we present Memformer, which takes advantage of a memory system to efficiently process long sequences with a linear time complexity and constant memory complexity. Along with Memformer, we introduce a new optimization scheme, Memory Replay Back-propagation, which enables training recurrent neural networks with large memory. Our model achieves strong perplexity results on WikiText-103. It is also flexible to a wide range of self-supervised learning tasks. With the infinite temporal range capability, we believe Memformer can spark interesting works in domains such as lifelong learning and memory-augmented meta-learning.

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
