# OpenReview forum: "Memformer: The Memory-Augmented Transformer"
_ICLR.cc/2021/Conference — Reject_

### Official Review · AnonReviewer3 · 2020-10-28
**Official Blind Review by reviewer3**

**Rating:** 6
**Confidence:** 3

**Review:**

This paper proposes a new style transformer with external memory, which is updated and used through an attention mechanism. They also propose a new algorithm to train the memory, Memory Replay Back-Propagation (MRBP). The memory consists of key-value pair data and is recurrently updated after the segment encoding. Through this memory, it can attend the past knowledge without the limitation of the maximum temporal range. The MRBP algorithm trains the memory through the local back-propagation of loss to reduce memory overhead.

Strengths:
1. They propose a new type of transformer augmented explicit memory, which contains the entire past knowledge, so it can attend the past knowledge without the computational overhead.
2. They also propose a new training algorithm, MRBP to update the memory with the local back-propagation of loss, which makes sense and they showed empirical results about how much this algorithm reduces memory overhead efficiently.
3. They evaluated their method with baselines in the aspect of the number of parameters and training speed and performance for various memory sizes and reported plentiful ablation study results too.

Weaknesses:
1. I think that section 2.1 requires an additional description: (e.g., in (a), what is the meaning of W_rE_{x_j}?, in (c), how W_k E_{x_j} can be the global content bias?)
2. This Memformer requires recurrent processing segment by segment, which can be another limitation on training or processing (can require more time than naive Transformer, which can process parallelly). I think that it is a fundamental weakness on this type of Transformer and if there is the comparison with naive Transformer (if the device can train it), it would be interesting for me (I don't think that it is a critical weakness).

The correctness of their claim and Clarity:

This paper is well written and looks correct except for some parts like requiring more descriptions or related to the previous method.

Additional feedback:

Thank you for submitting it. I enjoyed reading it.

However for me, section 2.1 is hard to understand, if there is more description, then it can be easier.

And, you mentioned Transformer XL has a` theoretical maximum temporal range, but by using the hidden states of the previous segment (the authors called a recurrent connection between the segments),  Transformer XL can access the past knowledge. Could you explain more the theoretical maximum temporal range?

This Memformer looks like the combination of the Transformer and Recurrent module through the explicit memory. Like RNN, it doesn't need incremental memory overhead as the sequence is going on through the memory, and like the Transformer, it can predict the output through the attention modules. As I mentioned in the weakness part, exploring the best combination of RNN and the Transformer can be interesting.

Minor things are
On page 6, (L x Nm + c x Ncm -> (L x Nm + c x Ncm)
On page 7, when increasing the memory size from 1 to 8 -> 128

---

> ### Author Response · Authors · 2020-11-13
> **Response**
>
> Q: I think that section 2.1 requires an additional description: (e.g., in (a), what is the meaning of W_rE_{x_j}?, in (c), how W_k E_{x_j} can be the global content bias?)
>
> Sorry for the confusion. (c) W_k E_{x_j} is the global content bias because it projects local token representations to the attention scores and it is only conditional to the content W_kE_{x_j} instead of the position information.
>
>
> Q: This Memformer requires recurrent processing segment by segment, which can be another limitation on training or processing (can require more time than naive Transformer, which can process parallelly). I think that it is a fundamental weakness on this type of Transformer and if there is the comparison with naive Transformer (if the device can train it), it would be interesting for me (I don't think that it is a critical weakness).
>
> Recurrent processing or RNN is not necessarily a limitation after the emergence of Transformer. Particularly, the computation cost of self-attention is O(n^2), but RNN is O(n). If the sequence is very long, a naive Transformer would be very inefficient and slow. In our case, when we increase the sequence size to 512 and larger, the memory consumption is bigger than 16GB.
>
>
> Q: However for me, section 2.1 is hard to understand, if there is more description, then it can be easier.
>
> Thank you for the suggestion. We will include more description for the simplified RPE in the future.
>
> Q: And, you mentioned Transformer XL has a` theoretical maximum temporal range, but by using the hidden states of the previous segment (the authors called a recurrent connection between the segments), Transformer XL can access the past knowledge. Could you explain more the theoretical maximum temporal range?
>
> Because of the unidirectional self-attention which forms a lower triangle attention mask, if there is only one layer, the token processed at time t can at most see the token at time t - L where L is the memory size. Thus, the maximum temporal range for one layer Transformer XL is L.
>
> When we add another layer, the previous token at time t - L can be computed from the lower layer and at most see the token at time t - L - L. Thus, if we have N layers, the maximum theoretical temporal range for Transformer XL is O(NxL).

---

### Official Review · AnonReviewer1 · 2020-10-28
**A good concept but lack experimentation**

**Rating:** 5
**Confidence:** 3

**Review:**

The paper is clearly written and explains the concept well. Keeping a memory part of transformers network enables Memformer to achieve O(n) complexity in time and O(1) in space. It is particularly helpful for handling long sequences.
The authors proposed Memory replay Backpropagation scheme which is more effective and efficient than gradient checkpointing.
The authors draw a fair comparison with long-sequence models (Transformer XL and Compressive Transformer) on Wikitext-103 dataset.
The concept seems to be a mixture of some work done in the past already. However, making it work and slot and cross attentions tricks seems to be a good choice.

Weakness:

1. Lack of experimentation: The authors compared other two transformer networks on just one dataset and presented their results with a hypothesis. The hypothesis is hard to be validated on just one task. The claims that the architecture can be scaled to any task is not justified with any empirical evidence.
2. In continuation with previous point, the idea usage and applicability is not justified. There are sparse transformers already like Longformer, BigBird that can process long documents too. The lack of experimentation doesnot validate the usefulness of the architecture.
3. The memory scaling seems to be a problem. Also clearly seen from the graph and the experiments. Effect of memory size should further improve the results which is not there. It is left for future work by the authors which doesnot go well with the hypothesis. Further, the restriction on the memory size is too small. With the advent of larger models, the usage of smaller memory seems implausible.

Minor comments:
* The multi task sampling ratio [0.6, 0.3, 0.1] during training is not justified. Is it randomly chosen or was different ratios tried?

---

> ### Author Response · Authors · 2020-11-13
> **Response**
>
> Q: Lack of experimentation: The authors compared other two transformer networks on just one dataset and presented their results with a hypothesis. The hypothesis is hard to be validated on just one task. The claims that the architecture can be scaled to any task is not justified with any empirical evidence.
>
> Language modeling is a very basic task in NLP. Most new structures designed for NLP tasks first tried out on language modelings, such as Transformer and Transformer-XL.
> If the models perform well on language modeling, they are likely to perform well too on other down-stream tasks. We plan to further test our model on other downstream applications, such as question answering and natural language inference.
>
>
> Q: In continuation with previous point, the idea usage and applicability is not justified. There are sparse transformers already like Longformer, BigBird that can process long documents too. The lack of experimentation does not validate the usefulness of the architecture.
>
> Sparse transformers such as Longformer, BigBird are in an orthogonal direction to our approach. They optimize the self-attention pattern to achieve better efficiency. It is possible to combine sparse attention to our approach, but it is not the main focus of the paper.
>
> Q: The memory scaling seems to be a problem. Also clearly seen from the graph and the experiments. Effect of memory size should further improve the results which is not there. It is left for future work by the authors which doesnot go well with the hypothesis. Further, the restriction on the memory size is too small. With the advent of larger models, the usage of smaller memory seems implausible.
>
> Memory scaling is indeed a limitation of our model. One reason might be that the average document length is 3,600, and a memory size of 32 is enough to compress all the information in the previous text.
>
>
> Q: The multi task sampling ratio [0.6, 0.3, 0.1] during training is not justified. Is it randomly chosen or were different ratios tried?
>
> We have tried several settings and found that this ratio yields slightly better results.

---

### Official Review · AnonReviewer4 · 2020-10-28
**Conceptually neat idea but practical usefulness is questionable**

**Rating:** 4
**Confidence:** 4

**Review:**

Summary:
This paper proposes an encoder-decoder memory-augmented language model, called Memformer. Similar to recurrent neural networks, the memory part is updated as a new input token comes in. The author presents a new optimization method, which is a variant of gradient checkpointing.

Reasons for score:
Contrary to the paper's argument, I am not sure that Memformer improves computational complexity or performance on long-range sequence modeling.

Concerns/Questions:
- Contributions of the newly proposed methods (increased model capacity from memory augmentation, specially designed attention, multi-task self-supervised learning, simplified relative position embedding) to the performance are mixed. A more thorough ablation study is required.
- To evaluate the effectiveness of Memformer in long-range sequence modeling, I would recommend exploring other settings than WikiTxt-103 with Transformer-XL (or Compressive Transformer). I suggest following experimental settings in the Longformer paper [1].
- I feel that the memory size of 32 is unsatisfactory. The author should provide a plausible explanation about why increasing memory size larger than 32 is not helpful. To add, Lample et al. propose product key memory using 512^2 memory slots [2].
- I am curious whether the Memformer using a memory size of 32 is really doing well in long-term modeling even though memory vectors can memorize information from multiple tokens.
- The setting of the main experiment is not valid. Comparison of Transformer-XL and Compressive Transformer with Memformer by reducing memory size (attention length) is not fair because they are designed to capture long-term context.
- The number of parameters in Table 1 is quite far from the values in the original Transformer-XL paper. Could you elaborate on it?
- Could you provide detail of speed measurement in Table 1?
- Can we try a memory replacement policy like LRU (Least Recently Used) via a sparse update?

Minor comment:
- The name of memory *cross* attention and memory *slot* attention is not intuitive.
- Mention of the text continuation task appears in Section 2.2.3 before the definition in Section 2.3.

[1] Iz Beltagy, Matthew E. Peters, and Arman Cohan. “Longformer: The Long-Document Transformer”
[2] Guillaume Lample, Alexandre Sablayrolles, Marc'Aurelio Ranzato, Ludovic Denoyer, and Hervé Jégou. “Large Memory Layers with Product Keys”

---

> ### Author Response · Authors · 2020-11-13
> **Response**
>
> Q: Contributions of the newly proposed methods (increased model capacity from memory augmentation, specially designed attention, multi-task self-supervised learning, simplified relative position embedding) to the performance are mixed. A more thorough ablation study is required.
>
> We performed an ablation study to understand each component’s contribution to the enhanced performance. First, in Table 1, the newly proposed memory component brings 2.26 performance gain, where all models apply simplified RPE and we control the computation and memory cost. Second, having multi-task improves the performance slightly by 0.15, but its main purpose is to avoid overfitting as shown in Figure 3(a). Finally, the relative position’s ablation study is shown in Table 2, and we apply it to all the baseline models.
>
> Q: I feel that the memory size of 32 is unsatisfactory. The author should provide a plausible explanation about why increasing memory size larger than 32 is not helpful. To add, Lample et al. propose product key memory using 512^2 memory slots [2].
>
> As shown in Figure 3(c), the performance does not increase much after the memory size is bigger than 32. One reason might be because the average length of the document is 3,600 tokens, which is enough to cover with 32 memory slots.
>
> Also, we want to clarify that our memory is fundamentally different from Lample et al.’s memory. In their work, 512^2 memory slots are used statically as weights for the model to memorize the entire dataset’s information. However, our memory slots only need to dynamically compress the previous text information in a document, which is way much smaller.
>
>
> Q: The setting of the main experiment is not valid. Comparison of Transformer-XL and Compressive Transformer with Memformer by reducing memory size (attention length) is not fair because they are designed to capture long-term context.
>
> In Transformer XL and Compressive Transformer, the attention length including memory is up to 1,600. Notice that the average length of Wikipedia articles is 3.6K tokens.
> The computation cost of self-attention is O(n^2). A large memory size would be very inefficient, and it is against the purpose of using memory to compress information.
>
>
> Q: The number of parameters in Table 1 is quite far from the values in the original Transformer-XL paper.
>
> We are using a smaller size Transformer XL. Due to limited computation resources, we set the number of layers to 12, and the hidden size to 512. Also, we use BPE with vocabulary size 50,265 instead of 267,735 in the original Transformer XL, which also contributes to a smaller number of parameters.
>
> Q: Could you provide details of speed measurement in Table 1?
>
> The speed is measured on how many text segments of size 64 are trained in a second. We tested on 4 Nvidia V100 16GB GPUs on GCP and the total training time is about 10 hours
>
> Q: Can we try a memory replacement policy like LRU (Least Recently Used) via a sparse update?
>
> It is an interesting idea, but Transformer XL’s memory is based on previously computed tokens’ hidden states. Replacing some of the tokens might cause structure inconsistency between the current computed hidden states and the memory hidden states. Also, as mentioned in Compressive Transformer Section 5.5, older memories are not accessed less frequently.

---

### Official Review · AnonReviewer2 · 2020-10-31
**This is very much a work in progress, results are not directly comparable with previous works**

**Rating:** 3
**Confidence:** 4

**Review:**

Summary:
The paper presents a new model for the task of language modeling especially suited for longer sequences. This new model dubbed as Memformer consists of Transformer encoder-decoder and a memory module to store the past information from the encoder outputs. The encoder bidirectionally attends to the immediate previous sequence/segment information and to the memory module, which is designed to capture useful information from the past history of the full sequence. The idea is that by bidirectionally attending simultaneously to the previous input segment and to a memory module, the decoder should be able to improve its generation capabilities.

Pros:
The motivation of the proposed model is interesting, which is to bidirectionally encode the past segment information and a memory module to capture rich signals from the entire history.

Cons:
- The main drawback of this paper is the lack of controlled experiments in the results section. Due to this, fair comparisons to the previous language modeling results in the literature is not possible and thus merits of the approach are neither convincing nor clear.
- To evaluate performance of the Memformer model, the authors present results on the Wikitext-103 dataset. However, to compare with the perplexity results from baseline models such as Transformer-XL and Compressive Transformer, the authors don't include the results from the original papers and instead re-compute it under simplified settings. However, to report progress in a widely studied task such as language modeling, it is not fair to not compare against highly-cited state-of-the-art results.
- In Section 3.1, it is mentioned that byte-pair encodings are used to represent words. Are the perplexity results also computed over BPE tokens? If so, then I feel this evaluation scheme is inconsistent with the standard language modeling evaluation of tokens which is done over words (or linguistic units).
- Although the results in Table 3.2 show that Memformer models obtains small performance gains in perplexity over baselines, but these results are rather marginal improvements and in the absence of statistical significance testing results, it can't be really understood if these are actual performance gains or due to randomness in the training process.
- As the core of the experimental results are on Wikitext-103, it is unclear if the experimental findings would generalize. Currently, the paper lacks comparisons on other datasets such as PG-19 from Compressive Transformers.
- The authors propose a multi-task training approach for language modeling. However, the motivation and benefits of why doing this is actually needed is not clearly illustrated and the perplexity gains seem to be small.
- In Section 2.1, the motivation of simplified relative position encoding is not presented.
- In paragraph 3, it is mentioned that due to uni-directional attention is Transformer-XL style language models, the memory may not have enough capacity to retain important information. However, this does not necessarily holds true for different tasks. Can the authors include a more detailed explanation or cite a prior work that illustrates this phenomenon?


Writing Issues:
- abstract 1st line: The mention of "remarkable accomplishments" is very vague in the context of applicability of Transformer models.
- compatible with other self-supervised tasks: compatibility does not necessarily imply that such language models would be useful in self-supervised tasks.
- Introduction section is not well-written. For example, the transition from first paragraph to second paragraph is rather abrupt, paragraph 3: Transformers and its followers: this is a very informal writing style, something that is not really suited when submitting to a publication.
- Sec 2.2.1: The first sentence in third paragraph - "Figure 1b is an assumed language model." is grammatically incorrect.

---

> ### Author Response · Authors · 2020-11-13
> **Response**
>
> Q: The main drawback of this paper is the lack of controlled experiments in the results section. Due to this, fair comparisons to the previous language modeling results in the literature is not possible and thus merits of the approach are neither convincing nor clear.
>
> We tried our best to provide a complete and fair comparison of our model against other popular related models under the constraints of similar computation costs. We control many different hyperparameters such as the number of layers, vocabulary, batch size, learning rate, etc. to ensure fair comparisons, which requires more effort than simply reporting the results. But due to computation cost constraints, we weren’t able to run all models with their largest versions.
>
> One of the advantages of our model is being more efficient. Without a controlled setting, it would be unfair to compare efficiency. For example, in Table 1, we show that our model with 76.21M parameters can defeat the Transformer XL with 80.35M. We also measure the speed and find that the Compressive Transformer is actually not very efficient compared to the Transformer XL.
>
> Also, In the original baselines papers, the reported results are based on a much larger memory size of 1,600, which is not efficient as the attention cost is O(n^2). Thus, we shrink the memory size to a reasonable number.
>
> Moreover, we believe that only simply reporting results from the original papers without careful analysis would make an approach less convincing and clear.
> We contribute not only to our method but also the previous baselines by giving a detailed analysis of them. As Compressive Transformer has not released any code, we implement it carefully to ensure a fair comparison. We welcome people to test our implementation of the baselines.
>
> Q: In Section 3.1, it is mentioned that byte-pair encodings are used to represent words. Are the perplexity results also computed over BPE tokens? If so, then I feel this evaluation scheme is inconsistent with the standard language modeling evaluation of tokens which is done over words (or linguistic units).
>
> Yes. In the original Transformer XL, the perplexity is directly computed from the final loss. However, since Transformer XL did not use BPE, some UNKs (unknown token) are also included. We believe that it makes the comparisons less convincing. Therefore, we apply the widely used BPE instead, and the setting is consistent in all experiments including baselines.
>
> Q: Although the results in Table 3.2 show that Memformer models obtains small performance gains in perplexity over baselines, but these results are rather marginal improvements and in the absence of statistical significance testing results, it can't be really understood if these are actual performance gains or due to randomness in the training process.
>
> In our experiment, we improved the results from 22.44 to 21.26. In the original baselines, Transformer XL only improved from 18.7 to 18.3, and Comprive Transformer improved from 18.1 to 17.1. We believe that the perplexity difference is unlikely due to randomness. Also notice that we set dropout to 0.0 to eliminate randomness as much as possible.
>
>
> Q: The authors propose a multi-task training approach for language modeling. However, the motivation and benefits of why doing this is actually needed is not clearly illustrated and the perplexity gains seem to be small.
>
> As we have mentioned in the paper, perplexity gain is not the purpose of multi-task training. If you check Figure 3(a), we show that multi-task learning helps to stabilize the training to avoid overfitting.
>
> Q: In Section 2.1, the motivation of simplified relative position encoding is not presented.
>
> Our model needs relative position encoding to ensure when doing text continuation, the input segment for the encoder can adjust to different lengths. Moreover, we find that the RPE in Transformer XL can be simplified to have better performance.
>
>
>
>
> Q: In paragraph 3, it is mentioned that due to uni-directional attention is Transformer-XL style language models, the memory may not have enough capacity to retain important information
>
> A similar analysis can be found in T5 [1] Table 2, where a unidirection language model only achieves 73.78 in GLUE, but an encoder-decoder can achieve 79.60. The finding is similar to our claim that unidirectional attention is limited. We tested the memory module on a unidirectional language model, but the performance is bad.
>
> [1] Colin Raffel et al., Exploring the Limits of Transfer Learning with a Unified Text-to-Text Transformer

---

### Author Response · Authors · 2020-11-13
**To all reviewers**

Thank all reviewers for their comments. We want to emphasize that our work is to propose a new structure, Memformer, that combines the advantage of RNN and Transformer. This is an exploratory work that points out a new direction for long sequences modeling. Memformer takes the advantage of Transformer(self-attention)’s capability on processing short sequences and RNN’s capability to be efficient on long sequences. Experiments show our model is performing better than Transformer-XL and Compressive Transformer on a popular language modeling task. But like all other models from the beginning, our work may have some limitations and is not applicable for different tasks for now, but we will continue to explore the idea of using a single unified memory to improve the language model’s performance.

---

### Decision · Program_Chairs · 2021-01-07
**Final Decision**

**Decision:**

Reject

**Comment:**

This paper introduces a new model, called Memformer, that combines the strength of transformer networks and recurrent neural networks. While the reviewers found the idea interesting, they also raised issues regarding the experimental section. In particular, they found the results unconvincing, because of weak baselines, non standard experimental settings (eg. using reporting perplexity results on BPE tokens), or evaluating on only one dataset. These concerns were not well addressed by the rebuttal. For these reasons, I recommend to reject the paper.